# SGD Algorithms based on Incomplete $U$-statistics: Large-Scale Minimization of Empirical Risk

**Guillaume Papa, Stéphan Clémençon**
LTCI, CNRS, Télécom ParisTech
Université Paris-Saclay, 75013 Paris, France
`first.last@telecom-paristech.fr`

**Aurélien Bellet**
Magnet Team, INRIA Lille - Nord Europe
59650 Villeneuve d'Ascq, France
`aurelien.bellet@inria.fr`

## Abstract

In many learning problems, ranging from clustering to ranking through metric learning, empirical estimates of the risk functional consist of an average over tuples (*e.g.*, pairs or triplets) of observations, rather than over individual observations. In this paper, we focus on how to best implement a stochastic approximation approach to solve such risk minimization problems. We argue that in the large-scale setting, gradient estimates should be obtained by sampling tuples of data points with replacement (*incomplete $U$-statistics*) instead of sampling data points without replacement (*complete $U$-statistics based on subsamples*). We develop a theoretical framework accounting for the substantial impact of this strategy on the generalization ability of the prediction model returned by the Stochastic Gradient Descent (SGD) algorithm. It reveals that the method we promote achieves a much better trade-off between statistical accuracy and computational cost. Beyond the rate bound analysis, experiments on AUC maximization and metric learning provide strong empirical evidence of the superiority of the proposed approach.

## 1 Introduction

In many machine learning problems, the statistical risk functional is an expectation over $d$-tuples ($d \geq 2$) of observations, rather than over individual points. This is the case in supervised metric learning [3], where one seeks to optimize a distance function such that it assigns smaller values to pairs of points with the same label than to those with different labels. Other popular examples include bipartite ranking (see [27] for instance), where the goal is to maximize the number of concordant pairs (*i.e.* AUC maximization), and more generally multi-partite ranking (*cf* [12]), as well as pairwise clustering (see [7]). Given a data sample, the most natural empirical risk estimate (which is known to have minimal variance among all unbiased estimates) is obtained by averaging over *all* tuples of observations and thus takes the form of a $U$-statistic (an average of *dependent* variables generalizing the means, see [19]). The Empirical Risk Minimization (ERM) principle, one of the main paradigms of statistical learning theory, has been extended to the case where the empirical risk of a prediction rule is a $U$-statistic [5], using concentration properties of $U$-processes (*i.e.* collections of $U$-statistics). The computation of the empirical risk is however numerically unfeasible in large and even moderate scale situations due to the exploding number of possible tuples.

In practice, the minimization of such empirical risk functionals is generally performed by means of stochastic optimization techniques such as Stochastic Gradient Descent (SGD), where at each iteration only a small number of randomly selected terms are used to compute an estimate of the gradient (see [27, 24, 16, 26] for instance). A drawback of the original SGD learning method, introduced in the case where empirical risk functionals are computed by summing over independent observations (sample mean statistics), is its slow convergence due to the variance of the gradient estimates, see [15]. This has recently motivated the development of a wide variety of SGD variants implementing a variance reduction method in order to improve convergence. Variance reduction is

achieved by occasionally computing the exact gradient (see SAG [18], SVRG [15], MISO [20] and SAGA [9] among others) or by means of nonuniform sampling schemes (see [21, 28] for instance). However, such ideas can hardly be applied to the case under study here: due to the overwhelming number of possible tuples, computing even a single exact gradient or maintaining a probability distribution over the set of all tuples is computationally unfeasible in general.

In this paper, we leverage the specific structure and statistical properties of the empirical risk functional when it is of the form of a $U$-statistic to design an efficient implementation of the SGD learning method. We study the performance of the following sampling scheme for the gradient estimation step involved in the SGD algorithm: drawing with replacement a set of tuples directly (in order to build an *incomplete $U$-statistic gradient estimate*), rather than drawing a subset of observations without replacement and forming all possible tuples based on these (the corresponding gradient estimate is then a *complete $U$-statistic based on a subsample*). While [6] has investigated maximal deviations between $U$-processes and their incomplete approximations, the performance analysis carried out in the present paper is inspired from [4] and involves both the optimization error of the SGD algorithm and the estimation error induced by the statistical finite-sample setting. We first provide non-asymptotic rate bounds and asymptotic convergence rates for the SGD procedure applied to the empirical minimization of a $U$-statistic. These results shed light on the impact of the conditional variance of the gradient estimators on the speed of convergence of SGD. We then derive a novel generalization bound which depends on the variance of the sampling strategies. This bound establishes the indisputable superiority of the incomplete $U$-statistic estimation approach over the complete variant in terms of the trade-off between statistical accuracy and computational cost. Our experimental results on AUC maximization and metric learning tasks on large-scale datasets are consistent with our theoretical findings and show that the use of the proposed sampling strategy can provide spectacular performance gains in practice. We conclude this paper with promising lines for future research, in particular regarding the trade-offs involved in a possible implementation of nonuniform sampling strategies to further improve convergence.

The rest of this paper is organized as follows. In Section 2, we briefly review the theory of $U$-statistics and their approximations, together with elementary notions of gradient-based stochastic approximation. Section 3 provides a detailed description of the SGD implementation we propose, along with a performance analysis conditional upon the data sample. In Section 4, based on these results, we derive a generalization bound based on a decomposition into optimization and estimation errors. Section 5 presents our numerical experiments, and we conclude in Section 6. Technical proofs are sketched in the Appendix, and further details can be found in the Supplementary Material.

## 2 Background and Problem Setup

Here and throughout, the indicator function of any event $\mathcal{E}$ is denoted by $\mathbb{I}\{\mathcal{E}\}$ and the variance of any square integrable r.v. $Z$ by $\sigma^2(Z)$.

### 2.1 $U$-statistics: Definition and Examples

Generalized $U$-statistics are extensions of standard sample mean statistics, as defined below.

**Definition 1.** *Let $K \geq 1$ and $(d_1, \ldots, d_K) \in \mathbb{N}^{*K}$. Let $\mathbf{X}_{\{1, \ldots, n_k\}} = (X_1^{(k)}, \ldots, X_{n_k}^{(k)})$, $1 \leq k \leq K$, be $K$ independent samples of sizes $n_k \geq d_k$ and composed of i.i.d. random variables taking their values in some measurable space $\mathcal{X}_k$ with distribution $F_k(dx)$ respectively. Let $H : \mathcal{X}_1^{d_1} \times \cdots \times \mathcal{X}_K^{d_K} \to \mathbb{R}$ be a measurable function, square integrable with respect to the probability distribution $\mu = F_1^{\otimes d_1} \otimes \cdots \otimes F_K^{\otimes d_K}$. Assume in addition (without loss of generality) that $H(\mathbf{x}^{(1)}, \ldots, \mathbf{x}^{(K)})$ is symmetric within each block of arguments $\mathbf{x}^{(k)}$ (valued in $\mathcal{X}_k^{d_k}$), $1 \leq k \leq K$. The generalized (or $K$-sample) $U$-statistic of degrees $(d_1, \ldots, d_K)$ with kernel $H$, is then defined as*

$$U_{\mathbf{n}}(H) = \frac{1}{\prod_{k=1}^{K} \binom{n_k}{d_k}} \sum_{I_1} \cdots \sum_{I_K} H\left(\mathbf{X}_{I_1}^{(1)}; \mathbf{X}_{I_2}^{(2)}; \ldots; \mathbf{X}_{I_K}^{(K)}\right), \tag{1}$$

*where $\mathbf{n} = (n_1, \ldots, n_K)$, the symbol $\sum_{I_1} \cdots \sum_{I_K}$ refers to summation over all elements of $\Lambda$, the set of the $\prod_{k=1}^{K} \binom{n_k}{d_k}$ index vectors $(I_1, \ldots, I_K)$, $I_k$ being a set of $d_k$ indexes $1 \leq i_1 < \ldots < i_{d_k} \leq n_k$ and $\mathbf{X}_{I_k}^{(k)} = (X_{i_1}^{(k)}, \ldots, X_{i_{d_k}}^{(k)})$ for $1 \leq k \leq K$.*

In the above definition, standard mean statistics correspond to the case where $K = 1 = d_1$. More generally when $K = 1$, $U_\mathbf{n}(H)$ is an average over all $d_1$-tuples of observations. Finally, $K \geq 2$ corresponds to the multi-sample situation where a $d_k$-tuple is used for each sample $k \in \{1, \ldots, K\}$.

The key property of the statistic (1) is that it has minimum variance among all unbiased estimates of

$$\mu(H) = \mathbb{E}\left[H\left(X_1^{(1)}, \ldots, X_{d_1}^{(1)}, \ldots, X_1^{(K)}, \ldots, X_{d_k}^{(K)}\right)\right] = \mathbb{E}\left[U_\mathbf{n}(H)\right].$$

One may refer to [19] for further results on the theory of $U$-statistics. In machine learning, generalized $U$-statistics are used as performance criteria in various problems, such as those listed below.

**Clustering.** Given a distance $D : \mathcal{X}_1 \times \mathcal{X}_1 \to \mathbb{R}_+$, the quality of a partition $\mathcal{P}$ of $\mathcal{X}_1$ with respect to the clustering of an i.i.d. sample $X_1, \ldots, X_n$ drawn from $F_1(dx)$ can be assessed through the *within cluster point scatter*:

$$\widehat{W}_n(\mathcal{P}) = \frac{2}{n(n-1)} \sum_{i<j} D(X_i, X_j) \cdot \sum_{\mathcal{C} \in \mathcal{P}} \mathbb{I}\left\{(X_i, X_j) \in \mathcal{C}^2\right\}. \tag{2}$$

It is a one sample $U$-statistic of degree 2 with kernel $H_\mathcal{P}(x, x') = D(x, x') \cdot \sum_{\mathcal{C} \in \mathcal{P}} \mathbb{I}\{(x, x') \in \mathcal{C}^2\}$.

**Multi-partite ranking.** Suppose that $K$ independent i.i.d. samples $X_1^{(k)}, \ldots, X_{n_k}^{(k)}$ with $n_k \geq 1$ and $1 \leq k \leq K$ on $\mathcal{X}_1 \subset \mathbb{R}^p$ have been observed. The accuracy of a scoring function $s : \mathcal{X}_1 \to \mathbb{R}$ with respect to the $K$-partite ranking is empirically estimated by the rate of concordant $K$-tuples (sometimes referred to as the *Volume Under the ROC Surface*):

$$\widehat{\text{VUS}}_n(s) = \frac{1}{n_1 \times \cdots \times n_K} \sum_{k=1}^{K} \sum_{i_k=1}^{n_k} \mathbb{I}\left\{s(X_{i_1}^{(1)}) < \cdots < s(X_{i_K}^{(K)})\right\}.$$

The quantity above is a $K$-sample $U$-statistic with degrees $d_1 = \ldots = d_K = 1$ and kernel $\bar{H}_s(x_1, \ldots, x_K) = \mathbb{I}\{s(x_1) < \cdots < s(x_K)\}$.

**Metric learning.** Based on an i.i.d. sample of labeled data $(X_1, Y_1), \ldots, (X_n, Y_n)$ on $\mathcal{X}_1 = \mathbb{R}^p \times \{1, \ldots, J\}$, the empirical pairwise classification performance of a distance $D : \mathcal{X}_1 \times \mathcal{X}_1 \to \mathbb{R}_+$ can be evaluated by:

$$\widehat{R}_n(D) = \frac{6}{n(n-1)(n-2)} \sum_{i<j<k} \mathbb{I}\left\{D(X_i, X_j) < D(X_i, X_k), Y_i = Y_j \neq Y_k\right\}, \tag{3}$$

which is a one sample $U$-statistic of degree three with kernel $\tilde{H}_D((x, y), (x', y'), (x'', y'')) = \mathbb{I}\{D(x, x') < D(x, x''), y = y' \neq y''\}$.

## 2.2 Gradient-based minimization of $U$-statistics

Let $\Theta \subset \mathbb{R}^q$ with $q \geq 1$ be some parameter space and consider the risk minimization problem $\min_{\theta \in \Theta} L(\theta)$ with

$$L(\theta) = \mathbb{E}[H(X_1^{(1)}, \ldots, X_{d_1}^{(1)}, \ldots, X_1^{(K)}, \ldots, X_{d_K}^{(K)}; \theta)] = \mu(H(.; \theta)),$$

where $H : \prod_{k=1}^{K} \mathcal{X}_k^{d_k} \times \Theta \to \mathbb{R}$ is a convex loss function, the $(X_1^{(k)}, \ldots, X_{d_k}^{(k)})$'s, $1 \leq k \leq K$, are $K$ independent random variables with distribution $F_k^{\otimes d_k}(dx)$ on $\mathcal{X}_k^{d_k}$ respectively so that $H$ is square integrable for any $\theta \in \Theta$. Based on $K$ independent i.i.d. samples $X_1^{(k)}, \ldots, X_{n_k}^{(k)}$ with $1 \leq k \leq K$, the empirical version of the risk function is $\theta \in \Theta \mapsto \widehat{L}_\mathbf{n}(\theta) = U_\mathbf{n}(H(.; \theta))$. We denote by $\nabla_\theta$ the gradient operator w.r.t. $\theta$.

Many learning algorithms are based on gradient descent, following the iterations $\theta_{t+1} = \theta_t - \gamma_t \nabla_\theta \widehat{L}_\mathbf{n}(\theta_t)$, with an arbitrary initial value $\theta_0 \in \Theta$ and a learning rate (step size) $\gamma_t \geq 0$ such that $\sum_{t=1}^{+\infty} \gamma_t = +\infty$ and $\sum_{t=1}^{+\infty} \gamma_t^2 < +\infty$. Here we place ourselves in a large-scale setting, where the sample sizes $n_1, \ldots, n_K$ of the training datasets are such that computing the empirical gradient

$$\widehat{g}_\mathbf{n}(\theta) \stackrel{def}{=} \nabla_\theta \widehat{L}_\mathbf{n}(\theta) = \left(1/\prod_{k=1}^{K}\binom{n_k}{d_k}\right) \sum_{I_1} \cdots \sum_{I_K} \nabla_\theta H(\mathbf{X}_{I_1}^{(1)}; \mathbf{X}_{I_2}^{(2)}; \ldots; \mathbf{X}_{I_K}^{(K)}; \theta) \tag{4}$$

at each iteration is intractable due to the number $\#\Lambda = \prod_{k=1}^{K}\binom{n_k}{d_k}$ of terms to be averaged. Instead, stochastic approximation suggests the use of an unbiased estimate of (4) that is cheap to compute.

## 3 SGD Implementation based on Incomplete $U$-Statistics

A possible approach consists in replacing (4) by a (complete) $U$-statistic computed from subsamples of reduced sizes $n'_k << n_k$, $\{(X_1^{\prime(k)}, \ldots, X_{n'_k}^{\prime(k)}) : k = 1, \ldots, K\}$ say, drawn uniformly at random without replacement among the original samples, leading to the following gradient estimator:

$$\tilde{g}_{\mathbf{n}'}(\theta) = \frac{1}{\prod_{k=1}^{K} \binom{n'_k}{d_k}} \sum_{I_1} \ldots \sum_{I_K} \nabla_\theta H(\mathbf{X}_{I_1}^{\prime(1)}; \mathbf{X}_{I_2}^{\prime(2)}; \ldots; \mathbf{X}_{I_K}^{\prime(K)}; \theta), \tag{5}$$

where $\sum_{I_k}$ refers to summation over all $\binom{n'_k}{d_k}$ subsets $\mathbf{X}'^{(k)}_{I_k} = (X'^{(k)}_{i_1}, \ldots, X'^{(k)}_{i_{d_k}})$ related to a set $I_k$ of $d_k$ indexes $1 \le i_1 < \ldots < i_{d_k} \le n'_k$ and $\mathbf{n}' = (n'_1, \ldots, n'_K)$. Although this approach is very natural, one can obtain a better estimate for the same computational cost, as shall be seen below.

### 3.1 Monte-Carlo Estimation of the Empirical Gradient

From a practical perspective, the alternative strategy we propose is of disarming simplicity. It is based on a Monte-Carlo sampling scheme that consists in drawing independently with replacement among the set of index vectors $\Lambda$, yielding a gradient estimator in the form of a so-called *incomplete* $U$-statistic (see [19]):

$$\bar{g}_B(\theta) = \frac{1}{B} \sum_{(I_1, \ldots, I_K) \in \mathcal{D}_B} \nabla_\theta H(\mathbf{X}_{I_1}^{(1)}, \ldots, \mathbf{X}_{I_K}^{(K)}; \theta), \tag{6}$$

where $\mathcal{D}_B$ is built by sampling $B$ times with replacement in the set $\Lambda$. We point out that the conditional expectation of (6) given the $K$ observed data samples is equal to $\widehat{g}_{\mathbf{n}}(\theta)$. The parameter $B$, corresponding to the number of terms to be averaged, controls the computational complexity of the SGD implementation. Observe incidentally that an incomplete $U$-statistic is not a $U$-statistic in general. Hence, as an unbiased estimator of the gradient of the statistical risk $L(\theta)$, (6) is of course less accurate than the full empirical gradient (4) (*i.e.*, it has larger variance), but this slight increase in variance leads to a large reduction in computational cost. In our subsequent analysis, we will show that for the same computational cost (*i.e.*, taking $B = \prod_{k=1}^{K} \binom{n'_k}{d_k}$), implementing SGD with (6) rather than (5) leads to much more accurate results. We will rely on the fact that (6) has smaller variance w.r.t. to $\nabla L(\theta)$ (except in the case where $K = 1 = d_1$), as shown in the proposition below.

**Proposition 1.** *Set $B = \prod_{k=1}^{K} \binom{n'_k}{d_k}$. There exists a universal constant $c > 0$, such that we have:*

$$\sigma^2\left(\tilde{g}_{\mathbf{n}'}(\theta)\right) \le c \cdot \sigma_\theta^2 / \sum_{k=1}^{K} n'_k \quad and \quad \sigma^2\left(\bar{g}_B(\theta)\right) \le c \cdot \sigma_\theta^2 / \prod_{k=1}^{K} \binom{n'_k}{d_k},$$

*for all $\mathbf{n} \in \mathbb{N}^{*K}$, with $\sigma_\theta^2 = \sigma^2(\nabla_\theta H(X_1^{(1)}, \ldots, X_{d_K}^{(K)}; \theta))$. Explicit but lengthy expressions of the variances are given in [19].*

**Remark 1.** *The results of this paper can be extended to other sampling schemes to approximate (4), such as Bernoulli sampling or sampling without replacement in $\Lambda$, following the proposal of [14]. For clarity, we focus on sampling with replacement, which is computationally more efficient.*

### 3.2 A Conditional Performance Analysis

As a first go, we investigate and compare the performance of the SGD methods described above conditionally upon the observed data samples. For simplicity, we denote by $\mathbb{P}_{\mathbf{n}}(.)$ the conditional probability measure given the data and by $\mathbb{E}_{\mathbf{n}}[.]$ the $\mathbb{P}_{\mathbf{n}}$-expectation. Given a matrix $M$, we denote by $M^T$ the transpose of $M$ and $\|M\|_{HS} := \sqrt{Tr(MM^T)}$ its Hilbert-Schmidt norm. We assume that the loss function $H$ is $l$-smooth in $\theta$, i.e its gradient is $l$-Lipschitz, with $l > 0$. We also restrict ourselves to the case where $\widehat{L}_{\mathbf{n}}$ is $\alpha$-strongly convex for some deterministic constant $\alpha$:

$$\widehat{L}_{\mathbf{n}}(\theta_1) - \widehat{L}_{\mathbf{n}}(\theta_2) \le \nabla_\theta \widehat{L}_{\mathbf{n}}(\theta_1)^T (x - y) - \frac{\alpha}{2} \|\theta_1 - \theta_2\|^2 \tag{7}$$

and we denote by $\theta_{\mathbf{n}}^*$ its unique minimizer. We point out that the present analysis can be extended to the smooth but non-strongly convex case, see [1]. A classical argument based on convex analysis and

stochastic optimization (see [1, 22] for instance) shows precisely how the conditional variance of the gradient estimator impacts the empirical performance of the solution produced by the corresponding SGD method and thus strongly advocates the use of the SGD variant proposed in Section 3.1.

**Proposition 2.** *Consider the recursion $\theta_{t+1} = \theta_t - \gamma_t g(\theta_t)$ where $\mathbb{E}_\mathbf{n}[g(\theta_t)|\theta_t] = \nabla_\theta \widehat{L}_\mathbf{n}(\theta_t)$, and denote by $\sigma_n^2(g(\theta))$ the conditional variance of $g(\theta)$. For step size $\gamma_t = \gamma_1/t^\beta$, the following holds.*

1. *If $\frac{1}{2} < \beta < 1$, then:*

$$\mathbb{E}_\mathbf{n}[\widehat{L}_\mathbf{n}(\theta_{t+1}) - \widehat{L}_\mathbf{n}(\theta_\mathbf{n}^*)] \leqslant \frac{\sigma_\mathbf{n}^2(g(\theta_\mathbf{n}^*))}{t^\beta} \underbrace{\gamma_1 l 2^{\beta-1}(\frac{1}{2\alpha} + \frac{l\gamma_1^2}{2\beta-1})}_{C_1} + o(\frac{1}{t^\beta}).$$

2. *If $\beta = 1$ and $\gamma_1 > \frac{1}{2\alpha}$, then:*

$$\mathbb{E}_\mathbf{n}[\widehat{L}_\mathbf{n}(\theta_{t+1}) - \widehat{L}_\mathbf{n}(\theta_\mathbf{n}^*)] \leqslant \frac{\sigma_\mathbf{n}^2(g(\theta_\mathbf{n}^*))}{t+1} \underbrace{\frac{2^{\alpha\gamma_1} l \exp(2\alpha l \gamma_1^2) \gamma_1^2}{(2\alpha\gamma_1 - 1)}}_{C_2} + o(\frac{1}{t}).$$

Proposition 2 illustrates the well-known fact that the convergence rate of SGD is dominated by the variance term and thus one needs to focus on reducing this term to improve its performance.

We are also interested in the asymptotic behavior of the algorithm (when $t \to +\infty$), under the following assumptions:

$\mathbf{A_1}$ The function $\widehat{L}_\mathbf{n}(\theta)$ is twice differentiable on a neighborhood of $\theta_\mathbf{n}^*$.

$\mathbf{A_2}$ The function $\nabla \widehat{L}_\mathbf{n}(\theta)$ is bounded.

Let us set $\Gamma = \nabla^2 \widehat{L}_n(\theta_\mathbf{n}^*)$. We establish the following result (refer to the Supplementary Material for a detailed proof).

**Theorem 1.** *Let the covariance matrix $\Sigma_\mathbf{n}^*$ be the unique solution of the Lyapunov equation:*

$$\Gamma\Sigma_\mathbf{n}^* + \Sigma_\mathbf{n}^*\Gamma - \eta\Sigma_\mathbf{n}^* = \Sigma_\mathbf{n}(\theta_\mathbf{n}^*), \tag{8}$$

*where $\Sigma_n(\theta_n^*) = \mathbb{E}_\mathbf{n}[g(\theta_n^*)g(\theta_n^*)^T]$ and $\eta = \gamma_1 > \frac{1}{2\alpha}$ if $\beta = 1$, 0 if not. Then, under Assumptions $\mathbf{A_1} - \mathbf{A_2}$, we have:*

$$1/\gamma_t \left( \widehat{L}_\mathbf{n}(\theta_t) - \widehat{L}_\mathbf{n}(\theta^*) \right) \Rightarrow \frac{1}{2}U^T(\Sigma_\mathbf{n}^*)^{1/2}\Gamma(\Sigma_\mathbf{n}^*)^{1/2}U,$$

*where $U \sim \mathcal{N}(0, I_q)$. In addition, in the case $\eta = 0$, we have :*

$$\|(\Sigma_\mathbf{n}^*\Gamma)^{1/2}\|_{HS}^2 = \mathbb{E}[U^T(\Sigma_\mathbf{n}^*)^{1/2}\Gamma(\Sigma_\mathbf{n}^*)^{1/2}U] = \frac{1}{2}\sigma_\mathbf{n}^2(g(\theta_\mathbf{n}^*)). \tag{9}$$

Theorem 1 reveals that the conditional variance term again plays a key role in the asymptotic performance of the algorithm. In particular, it is the dominating term in the precision of the solution. In the next section, we build on these results to derive a generalization bound in the spirit of [4] which explicitly depend on the true variance of the gradient estimator.

## 4 Generalization Bounds

Let $\theta^* = \text{argmin}_{\theta \in \Theta} L(\theta)$ be the minimizer of the true risk. As proposed in [4], the mean excess risk can be decomposed as follows: $\forall \mathbf{n} \in \mathbb{N}^{*K}$,

$$\mathbb{E}[L(\theta_t) - L(\theta^*)] \leq \underbrace{2\mathbb{E}\left[\sup_{\theta \in \Theta} |\widehat{L}_\mathbf{n}(\theta) - L(\theta)|\right]}_{\mathcal{E}_1} + \underbrace{\mathbb{E}\left[\widehat{L}_\mathbf{n}(\theta_t) - \widehat{L}_\mathbf{n}(\theta_\mathbf{n}^*)\right]}_{\mathcal{E}_2}. \tag{10}$$

Beyond the optimization error (the second term on the right hand side of (10)), the analysis of the generalization ability of the learning method previously described requires to control the estimation error (the first term). This can be achieved by means of the result stated below, which extends Corollary 3 in [5] to the $K$-sample situation.

**Proposition 3.** *Let $\mathcal{H}$ be a collection of bounded symmetric kernels on $\prod_{k=1}^{K} \mathcal{X}_k^{d_k}$ such that $\mathcal{M}_{\mathcal{H}} = \sup_{(H,x) \in \mathcal{H} \times \mathcal{X}} |H(x)| < +\infty$. Suppose also that $\mathcal{H}$ is a VC major class of functions with finite Vapnik-Chervonenkis dimension $V < +\infty$. Let $\kappa = \min\{\lfloor n_1/d_1 \rfloor, \ldots, \lfloor n_K/d_K \rfloor\}$. Then, for any $\mathbf{n} \in \mathbb{N}^{*K}$*

$$\mathbb{E}\left[\sup_{H \in \mathcal{H}} |U_{\mathbf{n}}(H) - \mu(H)|\right] \leq \mathcal{M}_{\mathcal{H}}\left\{2\sqrt{\frac{2V \log(1+\kappa)}{\kappa}}\right\}. \tag{11}$$

We are now ready to derive our main result.

**Theorem 2.** *Let $\theta_t$ be the sequence generated by SGD using the incomplete statistic gradient estimator (6) with $B = \prod_{k=1}^{K} \binom{n'_k}{d_k}$ terms for some $n'_1, \ldots, n'_K$. Assume that $\{L(.; \theta) : \theta \in \Theta\}$ is a VC major class class of finite VC dimension $V$ s.t.*

$$\mathcal{M}_{\Theta} = \sup_{\theta \in \Theta, (\mathbf{x}^{(1)}, \ldots, \mathbf{x}^{(K)}) \in \prod_{k=1}^{K} \mathcal{X}_k^{d_k}} |H(\mathbf{x}^{(1)}, \ldots, \mathbf{x}^{(K)}; \theta)| < +\infty, \tag{12}$$

*and $\mathcal{N}_{\Theta} = \sup_{\theta \in \Theta} \sigma_\theta^2 < +\infty$. If the step size satisfies the condition of Proposition 2, we have:*

$$\forall \mathbf{n} \in \mathbb{N}^{*K}, \quad \mathbb{E}[|L(\theta_t) - L(\theta^*)|] \leqslant \frac{C\mathcal{N}_{\Theta}}{Bt^\beta} + 2\mathcal{M}_{\Theta}\left\{2\sqrt{\frac{2V \log(1+\kappa)}{\kappa}}\right\}.$$

*For any $\delta \in (0,1)$, we also have with probability at least $1 - \delta$: $\forall \mathbf{n} \in \mathbb{N}^{*K}$,*

$$|L(\theta_t) - L(\theta^*)| \leqslant \left(\frac{C\mathcal{N}_{\Theta}}{Bt^\beta} + \sqrt{\frac{D_\beta \log(2/\delta)}{t^\beta}}\right) + 2\mathcal{M}_{\Theta}\left\{2\sqrt{\frac{2V \log(1+\kappa)}{\kappa}} + \sqrt{\frac{\log(4/\delta)}{\kappa}}\right\}. \tag{13}$$

*for some constants $C$ and $D_\beta$ depending on the parameters $l, \alpha, \gamma_1, a_1$.*

The generalization bound provided by Theorem 2 shows the advantage of using an incomplete $U$-statistic (6) as the gradient estimator. In particular, we can obtain results of the same form as Theorem 2 for the complete $U$-statistic estimator (5), but $B = \prod_{k=1}^{K} \binom{n'_k}{d_k}$ is then replaced by $\sum_{k=1}^{K} n'_k$ (following Proposition 1), leading to greatly damaged bounds. Using an incomplete $U$-statistic, we thus achieve better performance on the test set while reducing the number of iterations (and therefore the numbers of gradient computations) required to converge to a accurate solution. To the best of our knowledge, this is the first result of this type for empirical minimization of $U$-statistics. In the next section, we provide experiments showing that these gains are very significant in practice.

## 5 Numerical Experiments

In this section, we provide numerical experiments to compare the incomplete and complete $U$-statistic gradient estimators (5) and (6) in SGD when they rely on the same number of terms $B$. The datasets we use are available online.[1] In all experiments, we randomly split the data into 80% training set and 20% test set and sample 100K pairs from the test set to estimate the test performance. We used a step size of the form $\gamma_t = \gamma_1/t$, and the results below are with respect to the number of SGD iterations. Computational time comparisons can be found in the supplementary material.

**AUC Optimization**  We address the problem of learning a binary classifier by optimizing the Area Under the Curve, which corresponds to the VUS criterion (Eq. 2) when $K = 2$. Given a sequence of i.i.d observations $Z_i = (X_i, Y_i)$ where $X_i \in \mathbb{R}^p$ and $Y_i \in \{-1, 1\}$, we denote by $X^+ = \{X_i; Y_i = 1\}$, $X^- = \{X_i; Y_i = -1\}$ and $N = |X^+||X^-|$. As done in [27, 13], we take a linear scoring rule $s_\theta(x) = \theta^T x$ where $\theta \in \mathbb{R}^p$ is the parameter to learn, and use the logistic loss as a smooth convex function upper bounding the Heaviside function, leading to the following ERM problem:

$$\min_{\theta \in \mathbb{R}^p} \frac{1}{N} \sum_{X_i^+ \in X^+} \sum_{X_j^- \in X^-} \log(1 + \exp(s_\theta(X_i^-) - s_\theta(X_i^+))).$$

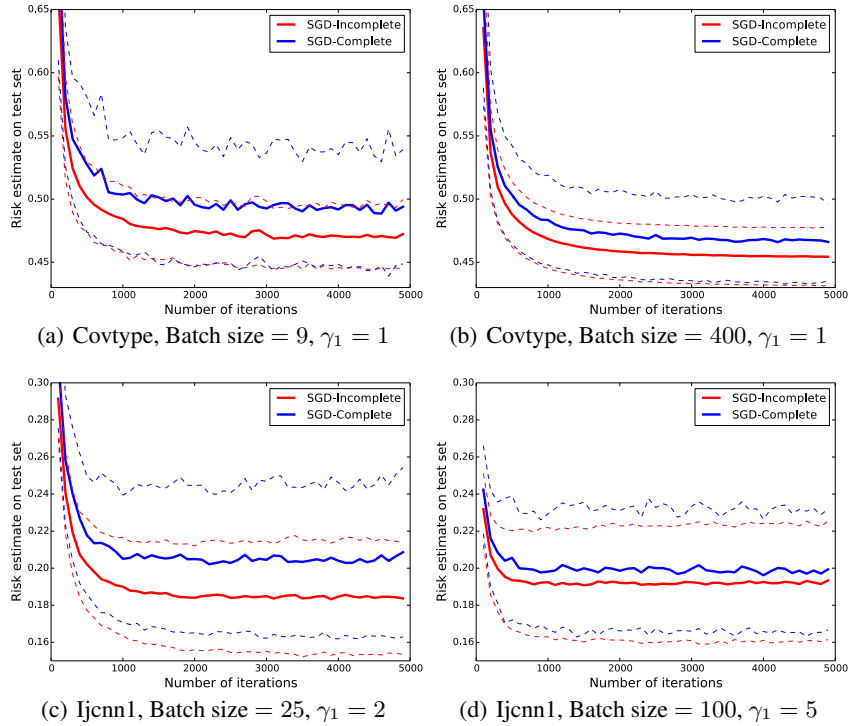

| (a) Covtype, Batch size $= 9$, $\gamma_1 = 1$ | (b) Covtype, Batch size $= 400$, $\gamma_1 = 1$ |
| (c) Ijcnn1, Batch size $= 25$, $\gamma_1 = 2$ | (d) Ijcnn1, Batch size $= 100$, $\gamma_1 = 5$ |

Figure 1: Average over 50 runs of the risk estimate with the number of iterations (solid lines) +/- their standard deviation (dashed lines)

We use two datasets: IJCNN1 ($\sim$200K examples, 22 features) and covtype ($\sim$600K examples, 54 features). We try different values for the initial step size $\gamma_1$ and the batch size $B$. Some results, averaged over 50 runs of SGD, are displayed in Figure 1. As predicted by our theoretical findings, we found that the incomplete $U$-statistic estimator always outperforms its complete variant. The performance gap between the two strategies can be small (for instance when $B$ is very large or $\gamma_1$ is unnecessarily small), but for values of the parameters that are relevant in practical scenarios (*i.e.*, $B$ reasonably small and $\gamma_1$ ensuring a significant decrease in the objective function), the difference can be substantial. We also observe a smaller variance between SGD runs with the incomplete version.

**Metric Learning** We now turn to a metric learning formulation, where we are given a sample of $N$ i.i.d observations $Z_i = (X_i, Y_i)$ where $X_i \in \mathbb{R}^p$ and $Y_i \in \{1, \ldots, c\}$. Following the existing literature [2], we focus on (pseudo) distances of the form $D_M(x, x') = (x - x')^T M (x - x')$ where $M$ is a $p \times p$ symmetric positive semi-definite matrix. We again use the logistic loss to obtain a convex and smooth surrogate for (3). The ERM problem is as follows:

$$\min_M \frac{6}{N(N-1)(N-2)} \sum_{i<j<k} \mathbb{I}\{Y_i = Y_j \neq Y_k\} \log(1 + \exp(D_M(X_i, X_j) - D_M(X_i, X_k))).$$

We use the binary classification dataset SUSY (5M examples, 18 features). Figure 2 shows that the performance gap between the two strategies is much larger on this problem. This is consistent with the theory: one can see from Proposition 1 that the variance gap between the incomplete and the complete approximations is much wider for a one-sample $U$-statistic of degree 3 (metric learning) than for a two-sample $U$-statistic of degree 1 (AUC optimization).

## 6    Conclusion and Perspectives

In this paper, we have studied a specific implementation of the SGD algorithm when the natural empirical estimates of the objective function are of the form of generalized $U$-statistics. This situation

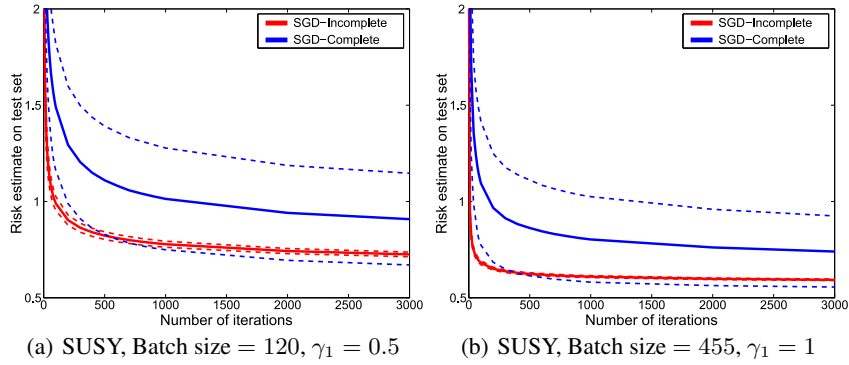

(a) SUSY, Batch size $= 120$, $\gamma_1 = 0.5$      (b) SUSY, Batch size $= 455$, $\gamma_1 = 1$

Figure 2: Average over 50 runs of the error test with the number of iterations (solid lines) +/- their standard deviation (dashed lines)

covers a wide variety of statistical learning problems such as multi-partite ranking, pairwise clustering and metric learning. The gradient estimator we propose in this context is based on an incomplete $U$-statistic obtained by sampling tuples with replacement. Our main result is a thorough analysis of the generalization ability of the predictive rules produced by this algorithm involving both the optimization and the estimation error in the spirit of [4]. This analysis shows that the SGD variant we propose far surpasses a more naive implementation (of same computational cost) based on subsampling the data points without replacement. Furthermore, we have shown that these performance gains are very significant in practice when dealing with large-scale datasets. In future work, we plan to investigate how one may extend the nonuniform sampling strategies proposed in [8, 21, 28] to our setting in order to further improve convergence. This is a challenging goal since we cannot hope to maintain a distribution over the set of all possible tuples of data points. A tractable solution could involve approximating the distribution in order to achieve a good trade-off between statistical performance and computational/memory costs.

## Appendix - Sketch of Technical Proofs

Note that the detailed proofs can be found in the Supplementary Material.

### Sketch Proof of Proposition 2

Set $a_t = \mathbb{E}_n[\|\theta_{t+1} - \theta_n^*\|^2]$ and following [1], observe that the sequence $(a_t)$ satisfies the recursion $a_{t+1} \leqslant a_t \left(1 - 2\alpha\gamma_t(1 - \gamma_t L)\right) + 2\gamma_t^2 \sigma_n^2(\theta_n^*)$. A standard stochastic approximation argument yields an upper bound for $a_t$ (cf [17, 1]), which, combined with $\widehat{L}_n(\theta) - \widehat{L}_n(\theta_n^*) \leqslant \frac{L}{2}\|\theta - \theta_n^*\|^2$ (see [23] for instance), give the desired result.

### Sketch of Proof of Theorem 1

The proof relies on stochastic approximation arguments (see [10, 25, 11]). We first show that $\sqrt{1/\gamma_t}\,(\theta_t - \theta_\mathbf{n}^*) \Rightarrow \mathcal{N}(0.\Sigma_\mathbf{n}^*)$. Then, we apply the second order delta-method to derive the asymptotic behavior of the objective function. Eq. (9) is obtained by standard algebra.

### Sketch of Proof of Theorem 2

Combining (10), (12) and Proposition 2 leads to the first part of the result. To derive sharp probability bounds, we apply the union bound on $\mathcal{E}_1 + \mathcal{E}_2$. To deal with $\mathcal{E}_1$, we use concentration results for $U$-processes, while we adapt the proof of Proposition 2 to control $\mathcal{E}_2$: the r.v.'s are recentered to make martingale increments appear, and finally we apply Azuma and Hoeffding inequalities.

**Acknowledgements** This work was supported by the chair Machine Learning for Big Data of Télécom ParisTech, and was conducted when A. Bellet was affiliated with Télécom ParisTech.

## Footnotes

[1] http://www.csie.ntu.edu.tw/~cjlin/libsvmtools/datasets/

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
