[Supplementary Material]

# SGD Algorithms based on Incomplete $U$-statistics: Large-Scale Minimization of Empirical Risk Supplementary Material

**Guillaume Papa, Stéphan Clémençon**
LTCI, CNRS, Télécom ParisTech
Université Paris-Saclay, 75013 Paris, France
first.last@telecom-paristech.fr

**Aurélien Bellet**
Magnet Team, INRIA Lille - Nord Europe
59650 Villeneuve d'Ascq, France
aurelien.bellet@inria.fr

## 1 Proof of Proposition 2

We follow the proof of [1] to derive bounds. We highlight the fact that since the loss function $H$ is $l$-smooth, $\widehat{g}_{\mathbf{n}}(\theta_t)$ and $\tilde{g}_{n'}(\theta_t)$ are $l$-Lipschitz. We introduce the sequence $\tilde{\gamma}_t = \gamma_t(1 - l\gamma_t)$. In all generality we will denote by $g_t(\theta)$ an unbiased estimator of the gradient at iteration $t$, $l$-Lipschitz in $\theta$. We study the recursion $\theta_{t+1} = \theta_t - \gamma_t g_t(\theta_t)$.

We will make use of the two following classical inequalities from convex analysis (see [7]) :

$$\widehat{L}_{\mathbf{n}}(\theta_1) - \widehat{L}_{\mathbf{n}}(\theta_2) \leqslant \nabla \widehat{L}_{\mathbf{n}}(\theta_1)^T (x - y) - \frac{\alpha}{2}\|\theta_1 - \theta_2\|^2 \tag{1}$$

$$\frac{1}{l}\|g_t(\theta_1) - g_t(\theta_2)\|^2 \leqslant (g_t(\theta_1) - g_t(\theta_1))^T (\theta_1 - \theta 2) \tag{2}$$

As mentioned previously the analysis we proposed can easily be extended to a more general setting as in [1]. We now begin the proof of the proposition :

$$\|\theta_{t+1} - \theta_n^*\|^2 = \|\theta_t - \theta_n^*\|^2 - 2\gamma_t g_t(\theta_t)^T(\theta_t - \theta_n^*) + \gamma_t^2 \|g_t(\theta_t)\|^2$$

Using (2) we get

$$\begin{aligned}\|g_t(\theta_t)\|^2 &\leqslant 2(\|g_t(\theta_t) - g_t(\theta_n^*)\|^2 + \|g_t(\theta_n^*)\|^2)\\ &\leqslant 2l(g_t(\theta_t) - g_t(\theta_n^*))^T(\theta_t - \theta_n^*) + 2\|g_t(\theta_n^*)\|^2)\end{aligned} \tag{3}$$

which together with $\mathbb{E}_{\mathbf{n}}[g_t(\theta_t)|\theta_t] = \widehat{g}_{\mathbf{n}}(\theta_t)$ gives

$$\mathbb{E}_{\mathbf{n}}[\|\theta_{t+1} - \theta_n^*\|^2|\theta_t] \leqslant \|\theta_t - \theta_n^*\|^2 - 2\tilde{\gamma}_t \widehat{g}_{\mathbf{n}}(\theta_t)^T(\theta_t - \theta) + 2\gamma_t^2 \|g_t(\theta_n^*)\|^2$$

For the sake of simplicity we assume $(1 - l\gamma_t) > 0 \,\forall t$ (which is eventually true since the sequence $(\gamma_t)_{t \geqslant 0}$ goes to 0 as $t$ goes to infinity). Let $a_t = \mathbb{E}_{\mathbf{n}}[\|\theta_t - \theta_n^*\|^2]$, $\sigma_n^2(\theta_n^*)$ the variance (conditionally upon the data) of $g_t(\theta_n^*)$. Using (1) and taking the expectation we get the following recursion :

$$\begin{aligned}a_{t+1} &\leqslant a_t \left(1 - 2\alpha\tilde{\gamma}_t\right) + 2\gamma_t^2 \sigma_n^2(\theta_n^*)\\ &\leqslant a_1 \prod_{j=1}^{t}\left(1 - 2\alpha\tilde{\gamma}_j\right) + 2\sigma_n^2(\theta_n^*)\sum_{j=1}^{t}\gamma_j^2 \prod_{k=j+1}^{t}\left(1 - 2\alpha\tilde{\gamma}_k\right)\end{aligned} \tag{4}$$

with the convention $\prod_{k=t+1}^{t}(1 - 2\alpha\tilde{\gamma}_k) = 1$. Using $1 + x \leqslant e^x$ we get the following upper bound :

$$\prod_{j=1}^{t}(1 - 2\alpha\tilde{\gamma}_j) \leqslant \exp(-2\alpha\sum_{j=1}^{t}\gamma_j)\exp(2\alpha l\sum_{j=1}^{t}\gamma_j^2)$$

We now need to distinguish two cases :

## 1.1 Case $\beta = 1$

If $\beta = 1$ we have :

1. $\log(t+1) - \log(j+1) \leqslant \sum_{k=j+1}^{t} \frac{1}{k}$

2. $\exp(2\alpha l \sum_{k=1}^{t} \frac{1}{k^2}) \leqslant \exp(4\alpha l)$

3. $\exp(2\alpha l \sum_{k=j+1}^{t} \frac{1}{k^2}) \leqslant \exp(\frac{2\alpha l}{j}) \leqslant \exp(2\alpha l)$

Under the assumption $2\alpha\gamma_1 > 1$ :

$$a_{t+1} \leqslant \frac{a_1}{(t+1)^{2\alpha\gamma_1}} \exp(4\alpha l\gamma_1^2) + 2\sigma_n^2(\theta_n^*)\exp(2\alpha l\gamma_1^2)\gamma_1^2 \sum_{j=1}^{t} \frac{(j+1)^{2\alpha\gamma_1}}{j^2} \frac{1}{(t+1)^{2\alpha\gamma_1}}$$

$$\leqslant \frac{a_1}{(t+1)^{2\alpha\gamma_1}} \exp(4\alpha l\gamma_1^2) + \frac{2^{\alpha\gamma_1} 2\sigma_n^2(\theta_n^*)\exp(2\alpha l\gamma_1^2)\gamma_1^2}{(2\alpha\gamma_1 - 1)(t+1)}$$

which gives the result.

## 1.2 Case $\beta < 1$

If $\beta < 1$, let $t_0$ be a positive index, by splitting the sum in two parts we get :

$$\sum_{j=1}^{t} \gamma_j^2 \prod_{k=j+1}^{t} (1 - 2\alpha\tilde{\gamma}_k) = \sum_{j=1}^{t_0} \gamma_j^2 \prod_{k=j+1}^{t} (1 - 2\alpha\tilde{\gamma}_k) + \sum_{j=t_0+1}^{t} \gamma_j^2 \prod_{k=j+1}^{t} (1 - 2\alpha\tilde{\gamma}_k)$$

$$\leqslant \prod_{k=t_0+1}^{t} (1 - 2\alpha\tilde{\gamma}_k) \sum_{j=1}^{t_0} \gamma_j^2 + \gamma_{t_0} \sum_{j=t_0+1}^{t} \gamma_j \prod_{k=j+1}^{t} (1 - 2\alpha\tilde{\gamma}_k)$$

where we used that the sequence $(\gamma_t)_{t \geqslant 1}$ is decreasing. Since $\gamma_j = \frac{1-(1-2\alpha\tilde{\gamma}_j)}{2\alpha} + l\gamma_j^2$ we have :

$$\sum_{j=t_0+1}^{t} \gamma_j \prod_{k=j+1}^{t} (1 - 2\alpha\tilde{\gamma}_k) = \frac{1}{2\alpha} \sum_{j=t_0+1}^{t} \prod_{k=j+1}^{t} (1 - 2\alpha\tilde{\gamma}_k) - \prod_{k=j}^{t} (1 - 2\alpha\tilde{\gamma}_k)$$

$$+ l \sum_{j=t_0+1}^{t} \gamma_j^2 \prod_{k=j+1}^{t} (1 - 2\alpha\tilde{\gamma}_k) \leqslant \frac{1}{2\alpha} + l \sum_{j=t_0+1}^{t} \gamma_j^2$$

which leads to

$$\sum_{j=1}^{t} \gamma_j^2 \prod_{k=j+1}^{t} (1 - 2\alpha\tilde{\gamma}_k) \leqslant \exp(-2\alpha \sum_{j=t_0+1}^{t} \gamma_j) \exp(2\alpha l \sum_{j=t_0+1}^{t} \gamma_j^2) \sum_{j=1}^{t_0} \gamma_j^2$$

$$+ \frac{\gamma_{t_0}}{2\alpha} + \gamma_{t_0} l \sum_{j=t_0+1}^{t} \gamma_j^2$$

Taking $t_0 \sim \frac{t}{2}$ and using the integral test for convergence :

1. $\sum_{j=t_0+1}^{t} \gamma_j = \gamma_1 \sum_{j=t_0+1}^{t} \frac{1}{j^\beta} \geqslant \gamma_1 \frac{(t+1)^{1-\beta} - (t_0+1)^{1-\beta}}{1-\beta} \geqslant \gamma_1 \frac{(t+1)^{1-\beta}}{2(1-\beta)}$

2. $\sum_{j=t_0+1}^{t} \gamma_j^2 = \gamma_1^2 \sum_{j=t_0+1}^{t} \frac{1}{j^{2\beta}} \leqslant \gamma_1^2 \sum_{j=2}^{+\infty} \frac{1}{j^{2\beta}} \leqslant \frac{\gamma_1^2}{2\beta-1}$

3. $\sum_{j=1}^{t_0} \gamma_j^2 \leqslant \gamma_1^2(1 + \frac{1}{2\beta-1}) = \frac{2\beta}{2\beta-1}\gamma_1^2$

gives the following bound :

$$a_{t+1} \leqslant a_1 \exp(-2\alpha\gamma_1 \frac{(t+1)^{1-\beta}}{2(1-\beta)}) \exp(2\alpha l \frac{\gamma_1^2}{2\beta-1})$$

$$+ 2\sigma_n^2(\theta_n^*)(\exp(-2\alpha\frac{(t+1)^{1-\beta}}{2(1-\beta)}) \exp(2\alpha l \frac{\gamma_1^2}{2\beta-1})\frac{2\beta}{2\beta-1}\gamma_1^2$$

$$+ \frac{2^\beta \gamma_1}{2\alpha t^\beta} + \frac{\gamma_1 2^\beta}{t^\beta}\frac{2l\beta}{2\beta-1}\gamma_1^2) = \sigma_n^2(\theta_n^*)\frac{\gamma_1 2^\beta}{t^\beta}(\frac{1}{2\alpha} + \frac{l\gamma_1^2}{2\beta-1}) + o(\frac{1}{t^\beta})$$

which concludes the proof.

## 2   Proof of Theorem 1

We recall that $\Gamma = \nabla^2 \widehat{L}_{\mathbf{n}}(\theta_n^*)$, $\Sigma_n(\theta_n^*) = \mathbb{E}_{\mathbf{n}}[g_t(\theta_n^*)g_t(\theta_n^*)^T]$ and $\Sigma_n^*$ is the solution of Lyapunov's equation :

$$\Gamma\Sigma_n^* + \Sigma_n^*\Gamma - \eta\Sigma_n^* = \Sigma_n(\theta_n^*), \tag{5}$$

Using classical results from stochastic approximation theory ( see [4, 5, 8] for instance) , we first show that under our assumptions:

$$\sqrt{1/\gamma_t}\,(\theta_t - \theta_n^*) \Rightarrow \mathcal{N}(0, \Sigma_n^*),$$

The asymptotic behavior of $1/\gamma_t\left(\widehat{L}_{\mathbf{n}}(\theta_t) - \widehat{L}_{\mathbf{n}}(\theta^*)\right)$ is therefore a consequence of the second order delta method. We now turn to the second part of proposition 3 and follow the analysis of [3]: We have

$$\mathbb{E}_{\mathbf{n}}[U^T(\Sigma_n^*)^{1/2}\Gamma(\Sigma_n^*)^{1/2}U] = \mathbb{E}_{\mathbf{n}}[Tr(\Gamma^{1/2}(\Sigma_n^*)^{1/2}UU^T(\Sigma_n^*)^{1/2}\Gamma^{1/2}]$$

$$= Tr(\Gamma^{1/2}(\Sigma_n^*)^{1/2}\mathbb{E}_{\mathbf{n}}[UU^T](\Sigma_n^*)^{1/2}\Gamma^{1/2})$$

$$= Tr(\Gamma^{1/2}(\Sigma_n^*)\Gamma^{1/2}) = Tr(\Gamma\Sigma_n^*)$$

$$= \frac{1}{2}Tr(\Sigma_n(\theta_n^*)) = \frac{1}{2}\sigma_n^2(\theta_n^*)$$

where we used the linearity of the trace, the linearity of the expectation, the dominated convergence theorem (to arrange the different terms) and Lyapunov's equation to conclude.

## 3   Proof of Theorem 2

We prove a more general result and apply it to our specific setting. We consider the recursion defined in the proof of Proposition 2 and keep the same notations.

**Theorem.** *Let $\theta_t$ be the sequence generated by SGD and define $\sigma^2 = \mathbb{E}[\sigma_{\mathbf{n}}^2(g(\theta_{\mathbf{n}}^*))]$. Assume that $\{L(.;\,\theta):\, \theta \in \Theta\}$ is a VC major class class of finite VC dimension $V$ s.t*

$$\mathcal{M}_\Theta = \sup_{\theta\in\Theta,\,(\mathbf{x}^{(1)},...,\mathbf{x}^{(K)})\in\prod_{k=1}^K \mathcal{X}_k^{d_k}} |H(\mathbf{x}^{(1)},\,...,\,\mathbf{x}^{(K)};\,\theta)| < +\infty, \tag{6}$$

*If the step size satisfies the condition of Proposition 2, we have:*

$$\forall \mathbf{n} \in \mathbb{N}^{*K}, \quad \mathbb{E}[|L(\theta_t) - L(\theta^*)|] \leqslant \frac{C\sigma^2}{t^\beta} + 2\mathcal{M}_\Theta\left\{2\sqrt{\frac{2V\log(1+\kappa)}{\kappa}}\right\}.$$

*For any $\delta \in (0,1)$, we also have with probability at least $1 - \delta$: $\forall \mathbf{n} \in \mathbb{N}^{*K}$,*

$$|L(\theta_t) - L(\theta^*)| \leqslant \left(\frac{C\sigma^2}{t^\beta} + \sqrt{\frac{D_\beta\log(2/\delta)}{t^\beta}}\right) + 2\mathcal{M}_\Theta\left\{2\sqrt{\frac{2V\log(1+\kappa)}{\kappa}} + \sqrt{\frac{\log(4/\delta)}{\kappa}}\right\}. \tag{7}$$

*for some constant $D_\beta$ depending on the parameters $l, \alpha, \gamma_1, a_1$ and where $C = C_1$ if $\beta < 1$ and $C_2$ otherwise.*

*Proof.* For the sake of simplicity, we place ourselves in the special case where $\Theta$ is compact, but tedious calculations would lead to similar results under less restrictive assumptions. We therefore introduce the quantities $M$ and $M_1$ that satisfy $\|g_t(\theta_n^*)\|^2 \leqslant M^2$ and $\|\theta_t - \theta_n^*\| \leqslant M_1^2$. We now turn to the proof of the result :

$$L(\theta_t) - L(\theta^*) \leq 2 \sup_{\theta \in \Theta} |\widehat{L}_{\mathbf{n}}(\theta) - L(\theta)| + \widehat{L}_{\mathbf{n}}(\theta_t) - \widehat{L}_{\mathbf{n}}(\theta_{\mathbf{n}}^*).$$

Taking a union bound we directly get :

$$\mathbb{P}\left( L(\theta_t) - L(\theta^*) \geqslant \frac{l}{2}\frac{\sigma^2}{t^\beta}C + \epsilon \right) \leqslant \underbrace{\mathbb{P}\left( |\widehat{L}_{\mathbf{n}}(\theta_t) - \widehat{L}_{\mathbf{n}}(\theta_n^*)| \geqslant \frac{l}{2}\frac{\sigma^2}{t^\beta}C + \frac{\epsilon}{2}) \right)}_{\mathcal{P}_1}$$

$$+ \underbrace{\mathbb{P}(\sup_{\theta \in \Theta}|\widehat{L}_{\mathbf{n}}(\theta) - L(\theta)| \geqslant \frac{\epsilon}{4})}_{\mathcal{P}_2}$$

The analysis of $\mathcal{P}_2$ is classical and we refer to [2] to obtain that for all $\delta \in (0,1)$, we have with probability at least $1 - \delta$,

$$\sup_{\theta \in \Theta}|\widehat{L}_{\mathbf{n}}(\theta) - L(\theta)| \leq \mathcal{M}_{\mathcal{H}}\left\{ 2\sqrt{\frac{2V\log(1+\kappa)}{\kappa}} + \sqrt{\frac{\log(1/\delta)}{\kappa}} \right\}. \tag{8}$$

We now focus on the second term.
Using $\widehat{L}_{\mathbf{n}}(\theta) - \widehat{L}_{\mathbf{n}}(\theta_n^*) \leqslant \frac{l}{2}\|\theta - \theta_n^*\|^2$ (see [6] for instance) we have :

$$\mathbb{P}\left( |\widehat{L}_{\mathbf{n}}(\theta_t) - \widehat{L}_{\mathbf{n}}(\theta_n^*)| - \frac{l}{2}\frac{\sigma^2}{t^\beta}C \geqslant \frac{\epsilon}{2} \right) \leqslant \mathbb{P}\left( \|\theta_t - \theta_n^*\|^2 - \frac{\sigma^2}{t^\beta}C \geqslant \frac{\epsilon}{l} \right)$$

Applying the recursion we get:
$$\|\theta_{t+1} - \theta_n^*\|^2 = \|\theta_t - \theta_n^*\|^2 - 2\gamma_t g_t(\theta_t)^T(\theta_t - \theta_n^*) + \gamma_t^2\|g_t(\theta_t)\|^2$$
$$= \|\theta_t - \theta_n^*\|^2 - 2\gamma_t(g_t(\theta_t) - \nabla\widehat{L}_{\mathbf{n}}(\theta_t))^T(\theta_t - \theta_n^*) - 2\gamma_t\nabla\widehat{L}_{\mathbf{n}}(\theta_t)^T(\theta_t - \theta_n^*) + \gamma_t^2\|g_t(\theta_t)\|^2$$

and using (3) :
$$\|g_t(\theta_t)\|^2 \leqslant 2l(g_t(\theta_t) - g_t(\theta_n^*))^T(\theta_t - \theta_n^*) + 2\|g_t(\theta_n^*)\|^2)$$
$$= 2l(g_t(\theta_t) - \nabla\widehat{L}_{\mathbf{n}}(\theta_t) - g_t(\theta_n^*))^T(\theta_t - \theta_n^*) + 2l(\nabla\widehat{L}_{\mathbf{n}}(\theta_t))^T(\theta_t - \theta_n^*) + 2\|g_t(\theta_n^*)\|^2$$

which with $\tilde{a}_t := \|\theta_{t+1} - \theta_n^*\|^2$ gives

$$\tilde{a}_{t+1} \leqslant \tilde{a}_t(1 - 2\alpha\tilde{\gamma}_t) + 2\gamma_t^2\sigma_n^2(\theta_n^*) - 2\tilde{\gamma}_t(g_t(\theta_t) - \nabla\widehat{L}_{\mathbf{n}}(\theta_t))^T(\theta_t - \theta_n^*)$$
$$- 2\gamma_t^2 l(g_t(\theta_n^*))^T(\theta_t - \theta_n^*) + 2\gamma_t^2(\|g_t(\theta_n^*)\|^2 - \sigma_n^2(\theta_n^*))).$$

An immediate recursion leads to

$$\tilde{a}_{t+1} \leqslant \tilde{a}_1 \prod_{j=1}^{t}(1 - 2\alpha\tilde{\gamma}_j) + 2\sigma_n^2(\theta_n^*)\sum_{j=1}^{t}\gamma_j^2\prod_{k=j+1}^{t}(1 - 2\alpha\tilde{\gamma}_k)$$

$$+ 2\sum_{j=1}^{t}\gamma_j^2\prod_{k=j+1}^{t}(1 - 2\alpha\tilde{\gamma}_k)\left(\|g_t(\theta_n^*)\|^2 - \sigma_n^2(\theta_n^*)\right)$$

$$- 2\sum_{j=1}^{t}\tilde{\gamma}_j\prod_{k=j+1}^{t}(1 - 2\alpha\tilde{\gamma}_k)\left(g_t(\theta_t) - \nabla\widehat{L}_{\mathbf{n}}(\theta_t)\right)^T(\theta_t - \theta_n^*)$$

$$- 2l\sum_{j=1}^{t}\gamma_j^2\prod_{k=j+1}^{t}(1 - 2\alpha\tilde{\gamma}_k)g_t(\theta_n^*)^T(\theta_t - \theta_n^*)$$

The first two terms are analyzed in Section 1. We turn now to the tree remaining terms and we introduce the following quantities :

1. $S_{1,t} = 2 \sum_{j=1}^{t} \gamma_j^2 \prod_{k=j+1}^{t} (1 - 2\alpha\tilde{\gamma}_k) \left( \|g_t(\theta_n^*)\|^2 - \sigma_n^2(\theta_n^*) \right)$

2. $S_{2,t} = 2 \sum_{j=1}^{t} \tilde{\gamma}_j \prod_{k=j+1}^{t} (1 - 2\alpha\tilde{\gamma}_k) (g_t(\theta_t) - \nabla\widehat{l_{\mathbf{n}}}(\theta_t))^T (\theta_t - \theta_n^*)$

3. $S_{3,t} = 2l \sum_{j=1}^{t} \gamma_j^2 \prod_{k=j+1}^{t} (1 - 2\alpha\tilde{\gamma}_k) g_t(\theta_n^*)^T (\theta_t - \theta_n^*)$

Placing ourself under the conditional probability and applying a union bound yields

$$\mathbb{P}_{\mathbf{n}} \left( \|\theta_{t+1} - \theta_n^*\|^2 \geqslant \tilde{a}_1 \prod_{j=1}^{t} (1 - 2\alpha\tilde{\gamma}_j) + 2\sigma_n^2(\theta_n^*) \sum_{j=1}^{t} \gamma_j^2 \prod_{k=j+1}^{t} (1 - 2\alpha\tilde{\gamma}_k) + \epsilon \right)$$

$$\leqslant \mathbb{P}_{\mathbf{n}}(S_{1,t} \geqslant \frac{\epsilon}{3}) + \mathbb{P}_{\mathbf{n}}(S_{2,t} \geqslant \frac{\epsilon}{3}) + \mathbb{P}_{\mathbf{n}}(S_{3,t} \geqslant \frac{\epsilon}{3})$$

Under our assumptions, we have $\|g_t(\theta_n^*)\|^2 \leqslant M^2$, $|g_t(\theta_n^*)^T (\theta_t - \theta_n^*)| \leqslant MM_1$ and $|(g_t(\theta_t) - \nabla\widehat{L_{\mathbf{n}}}(\theta_t))^T (\theta_t - \theta_n^*)| \leqslant (2lM_1 + M)M_1$. Applying Azuma's inequality yields the following bounds:

$$\mathbb{P}_{\mathbf{n}}(S_{1,t} \geqslant \epsilon) \leqslant \exp\left( \frac{-\epsilon^2}{4M^4 \sum_{j=1}^{t} \gamma_j^4 \prod_{k=j+1}^{t} (1 - 2\alpha\tilde{\gamma}_k)^2} \right)$$

$$\mathbb{P}_{\mathbf{n}}(S_{2,t} \geqslant \epsilon) \leqslant \exp\left( \frac{-\epsilon^2}{8M_1^2(2lM_1 + M)^2 \sum_{j=1}^{t} \tilde{\gamma}_j^2 \prod_{k=j+1}^{t} (1 - 2\alpha\tilde{\gamma}_k)^2} \right)$$

$$\mathbb{P}_{\mathbf{n}}(S_{3,t} \geqslant \epsilon) \leqslant \exp\left( \frac{-\epsilon^2}{8M^2 M_1^2 \sum_{j=1}^{t} \gamma_j^4 \prod_{k=j+1}^{t} (1 - 2\alpha\tilde{\gamma}_k)^2} \right)$$

We thus need to bound the sums $\sum_{j=1}^{t} \tilde{\gamma}_j^2 \prod_{k=j+1}^{t} (1 - 2\alpha\tilde{\gamma}_k)^2$ and $\sum_{j=1}^{t} \tilde{\gamma}_j^2 \prod_{k=j+1}^{t} (1 - 2\alpha\tilde{\gamma}_k)^2$.

## 3.1 Case $\beta < 1$

We have for $\beta < 1$:

$$\sum_{t=1}^{T} \gamma_j^4 \prod_{k=j+1}^{T} (1 - 2\alpha\tilde{\gamma}_k)^2 \leqslant \prod_{j=t_0+1}^{T} (1 - 2\alpha\tilde{\gamma}_j)^2 \sum_{j=1}^{t_0} \gamma_j^4 + \gamma_{t_0}^2 \sum_{j=t_0+1}^{T} \gamma_j^2 \prod_{k=j+1}^{t} (1 - 2\alpha\tilde{\gamma}_k)^2$$

$$\leqslant \exp(-4\alpha \sum_{j=t_0+1}^{T} \gamma_j) \exp(4\alpha l \sum_{j=t_0+1}^{T} \gamma_j^2) \sum_{j=1}^{t_0} \gamma_j^4$$

$$+ \gamma_{t_0}^2 (\frac{1}{2\alpha} + l \sum_{j=t_0+1}^{T} \gamma_j^2)^2$$

$$\leqslant \exp(-4\alpha\gamma_1 \frac{(T+1)^{(1-\beta)}}{2(1-\beta)}) \exp(\frac{4\alpha l\gamma_1^2}{2\beta - 1}) \frac{4\beta}{4\beta - 1} \gamma_1^4$$

$$+ \frac{\gamma_1^2 2^{2\beta}}{T^{2\beta}} (\frac{1}{2\alpha} + \frac{2l\beta}{2\beta - 1})^2$$

$$= \frac{\gamma_1^2 2^{2\beta}}{T^{2\beta}} (\frac{1}{2\alpha} + \frac{2l\beta}{2\beta - 1})^2 + o(\frac{1}{T^{2\beta}})$$

where we used $\sum_{j=1}^{T} x_j^2 \leqslant (\sum_{j=1}^{T} x_j)^2$ for $x_j = \gamma_j \prod_{k=j+1}^{t}(1 - 2\alpha\tilde{\gamma}_k) \geqslant 0$ for the last inequality. We now analyze the second sum :

$$\sum_{j=1}^{T} \tilde{\gamma}_j^2 \prod_{k=j+1}^{T}(1 - 2\alpha\tilde{\gamma}_k)^2 \leqslant \prod_{j=t_0+1}^{T}(1 - 2\alpha\tilde{\gamma}_j)^2 \sum_{j=1}^{t_0} \tilde{\gamma}_j^2 + \tilde{\gamma}_{t_0} \sum_{j=t_0+1}^{T} \tilde{\gamma}_j \prod_{k=j+1}^{T}(1 - 2\alpha\tilde{\gamma}_k)^2$$

$$\leqslant \exp(-4\alpha \sum_{j=t_0+1}^{T} \gamma_j)\exp(4\alpha l \sum_{j=t_0+1}^{T} \gamma_j^2)\sum_{j=1}^{t_0} \tilde{\gamma}_j^2$$

$$+ \tilde{\gamma}_{t_0} \sum_{j=t_0+1}^{T} \frac{1 - (1 - 2\alpha\tilde{\gamma}_j)}{2\alpha} \prod_{k=j+1}^{T}(1 - 2\alpha\tilde{\gamma}_k)$$

$$\leqslant \exp(-4\alpha \sum_{j=t_0+1}^{T} \gamma_j)\exp(4\alpha l \sum_{j=t_0+1}^{T} \gamma_j^2)\sum_{j=1}^{t_0} \tilde{\gamma}_j^2 + \frac{\tilde{\gamma}_{t_0}}{2\alpha}$$

$$\leqslant \exp(-4\alpha\gamma_1 \frac{(T+1)^{(1-\beta)}}{2(1-\beta)})\exp(\frac{4\alpha l\gamma_1^2}{2\beta - 1})\frac{8\beta}{2\beta - 1}\gamma_1^2 + \frac{\gamma_1 2^\beta}{\alpha T^\beta}$$

$$= \frac{\gamma_1 2^\beta}{\alpha T^\beta} + o(\frac{1}{T^\beta})$$

which concludes this case.

### 3.2  Case $\beta = 1$

We have:

$$\sum_{t=1}^{T} \gamma_j^4 \prod_{k=j+1}^{T}(1 - 2\alpha\tilde{\gamma}_k)^2 \leqslant \sum_{t=1}^{T} \gamma_j^4 \exp(-4\alpha \sum_{j=t+1}^{T} \gamma_j)\exp(4\alpha l \sum_{j=t+1}^{T} \gamma_j^2)$$

$$\leqslant \gamma_1^4 \exp(8\alpha l\gamma_1^2)\sum_{t=1}^{T} \frac{(j+1)^{4\alpha\gamma_1}}{j^4(T+1)^{4\alpha\gamma_1}}$$

$$\leqslant \frac{\gamma_1^4 \exp(8\alpha l\gamma_1^2)2^{4\alpha\gamma_1}}{(T+1)^{4\alpha\gamma_1}}\sum_{t=1}^{T} \frac{1}{j^{4-4\alpha\gamma_1}}$$

We thus need to distinguish several cases:

1. If $2 < 4\alpha\gamma_1 < 3$ then

$$\sum_{t=1}^{T} \frac{1}{j^{4-4\alpha\gamma_1}} \leqslant 1 + \frac{1}{3 - 4\alpha\gamma_1} = \frac{4 - 4\alpha\gamma_1}{3 - 4\alpha\gamma_1}$$

so

$$\sum_{t=1}^{T} \gamma_j^4 \prod_{k=j+1}^{T}(1 - 2\alpha\tilde{\gamma}_k)^2 \leqslant \frac{\gamma_1^4 \exp(8\alpha l\gamma_1^2)2^{4\alpha\gamma_1}}{(T+1)^{4\alpha\gamma_1}}\frac{4 - 4\alpha\gamma_1}{3 - 4\alpha\gamma_1}$$

2. $4\alpha\gamma_1 = 3$ then

$$\sum_{t=1}^{T} \frac{1}{j^{4-4\alpha\gamma_1}} \leqslant 1 + \log(T)$$

so

$$\sum_{t=1}^{T} \gamma_j^4 \prod_{k=j+1}^{T}(1 - 2\alpha\tilde{\gamma}_k)^2 \leqslant \gamma_1^4 \exp(8\alpha l\gamma_1^2)2^3 \frac{1 + \log(T)}{(T+1)^3}$$

3. $4\alpha\gamma_1 > 3$ then

$$\sum_{t=1}^{T} \frac{1}{j^{4-4\alpha\gamma_1}} \leqslant \frac{(T+1)^{4\alpha\gamma_1-3}}{4\alpha\gamma_1 - 3}$$

so

$$\sum_{t=1}^{T} \gamma_j^4 \prod_{k=j+1}^{T} (1 - 2\alpha\tilde{\gamma}_k)^2 \leqslant \frac{\gamma_1^4 \exp(8\alpha l \gamma_1^2) 2^{4\alpha\gamma_1}}{(4\alpha\gamma_1 - 3)(T+1)^3}$$

Consider $\gamma_1 < \frac{1}{2L}$, then $\tilde{\gamma}_j < \frac{1}{2}\gamma_j$, and

$$\sum_{j=1}^{t} \tilde{\gamma}_j^2 \prod_{k=j+1}^{T} (1 - 2\alpha\tilde{\gamma}_k)^2 \leqslant 4\gamma_1^2 \sum_{j=1}^{t} \frac{1}{j^2} \exp(-4\alpha \sum_{j=t+1}^{T} \gamma_j) \exp(4\alpha l \sum_{j=t+1}^{T} \gamma_j^2)$$

$$\leqslant \frac{4\gamma_1^2 \exp(8\alpha l \gamma_1^2) 2^{4\alpha\gamma_1}}{(T+1)^{4\alpha\gamma_1}} \sum_{t=1}^{T} \frac{1}{j^{2-4\alpha\gamma_1}}$$

$$\leqslant \frac{4\gamma_1^2 \exp(8\alpha l \gamma_1^2) 2^{4\alpha\gamma_1}}{(4\alpha\gamma_1 - 1)(T+1)}$$

bringing all the pieces back together and substituting the corresponding bounds give the result under the conditional probability $\mathbb{P}_{\mathbf{n}}$. Taking the expectation over the distribution of the datas gives the result in terms of $\mathbb{E}[\sigma_n^2(\theta_n^*)]$. Since $\mathbb{E}_{\mathbf{n}}[g(\theta_n^*)] = 0$, we have $\mathbb{E}[\sigma_n^2(\theta_n^*)] = \mathbb{E}[\|g(\theta_{\mathbf{n}}^*)\|^2] = \sigma^2(g(\theta_n^*))$ and we get the final result.

$\square$

# 4 Additional Experimental Results

In the main text, we provided results in terms of the number of iterations. For completeness, Figure 1 shows the corresponding results with respect to the computational time. The same conclusions hold regarding the relative performance of the two approaches.

(a) Covtype , Batch size $= 9, \gamma_1 = 1$

(b) Covtype , Batch size $= 400, \gamma_1 = 1$

(c) Ijcnn1 , Batch size $= 25, \gamma_1 = 2$

(d) Ijcnn1, Batch size $= 100, \gamma_1 = 5$

(e) SUSY, Batch size $= 120, \gamma_1 = 0.5$

(f) SUSY, Batch size $= 455, \gamma_1 = 1$

Figure 1: Average over 50 runs of risk estimate in terms of computational time in seconds (solid lines) +/- their standard deviation (dashed lines)