[Reviews · NeurIPS 2015]

Submitted by Assigned_Reviewer_1

The authors proposed to approximate the gradients for risk

minimization in large-scale settings using incomplete U-statistics, i.e., sampling tuples of data points with replacement. Previous work estimated the gradients by sampling data points without replacement (complete U-statistics based on subsamples), which is not efficient compared with the proposed incomplete U-statistics. The authors first provide the non-asymptotic rate bounds and asymptotic convergence rates for the SGD procedure applied to the empirical minimization of a U-statistics. A novel generalization bound is provided by the authors, from which we

can see that by using an incomplete U-statistics, we can achieve better performance for the same computational cost compared to a complete U-statistics. Finally, numerical experiments were conducted and the results showed the effectiveness of the proposed

method.

The quality of this paper is good. The authors try to achieve a good trade-off between statistical accuracy and computational cost for the well-known Empirical Risk Minimization problem in large-scale settings. The method proposed is easy to reproduce and the

datasets in the experiments are publicly available as well. One point which needs to be pointed is that all the bounds are based on

the assumption of the step size \gamma_t = \gamma_1/ t^{\beta}. What if the step size is not in this form? Besides, the step size in

the first experiment is in the form "\gamma_t = \frac{\gamma_1}{1+\gamma_1 \lambda t}, which is a different form. The paper should address this issue.

Overall, the paper is well written and easy to follow. The main contributions and technical contents are clearly stated. There are

two minor points that the paper could be better improved: a) It is a little confusing of the statement "Increasing slightly the variance of the gradient estimators is the price to pay for reducing SGD cost complexity". It is shown that (6) has smaller variance

than (5); however, the computational cost of (6) is also less than (5). b) In the experiment part, it could be better to elaborate how the batch size and the initial \gamma affect the performance, i.e., keeping one parameter fixed and observing the performance

changing with the other one.

The originality of this paper is good. In order to deal with large-scale minimization of empirical risk, the incomplete U-statistics are proposed, which achieves a trade-off between statistical accuracy and computational cost. The authors proposed to use Monte-Carlo method to estimate the empirical gradient, which produced better performance than using a U-statistic with subsamples. The technical proof is provided as well to show that incomplete U-statistics can achieve better performance while reducing the computation cost. This paper is of good significance to the community. As the data size is very large in the real world, we need to figure out how to efficiently solve ERM problem. The method proposed in this paper is easy to implement and has the potential to be adopted in both academia and industry.

In summary, the paper elegantly proposed to use incomplete U-statistics to deal with ERM problem in large-scale. A novel generalization bound is described to show that incomplete U-statistics can achieve better performance that complete U-statistics.
Summary: In this paper, the authors proposed to obtain the gradient estimates by incomplete U-statistics instead of complete U-statistics for the Empirical Risk Minimization in large-scale settings. A novel generalization bound was derived and results on numerical experiments show the effectiveness of the proposed method.

Submitted by Assigned_Reviewer_2

-- Review after rebuttal -- I thank the authors for their responses. I notched up my recommendation a bit, but still I cannot offer a clear accept for this interesting paper. My quibble with Prop. 1 was not fully resolved. The authors said "the variances \sigma are w.r.t. the gradient of the *true* loss L(\theta)". This is a bit unclear but I take it to mean that

\theta should be \theta^* in Prop. 1?

That would be an important typo that was not acknowledged. Furthermore, in later search through the citation the authors provide and other works, I found an inconsistency in the results, which basically boils down to the \sum n'_k being a product

in Prop. 1. Unfortunately, I cannot disclose other works to maintain author anonymity. Together with the way the authors originally presented this result, this makes me skeptical

about Prop. 1. Regarding my comments on excess mathematical clutter, these were generally acknowledged by the authors but it is unclear how much can be done in such limited time.

Still, I decided to increase my rating for this paper, so if other reviewers are happy with the

rebuttal the paper could get through.

--

In this paper the authors investigate ways to perform stochastic approximation that use U-statistics in a way that is computationally and statistically efficient. The quite surprising result is that using "incomplete U-statistics" is significantly

more efficient than "complete U-statistics", in both efficiency aspects. This is surprising because complete U-statistics use averages over all

"data tuples", albeit reducing the sample size.

I am skeptical about this result, which begins in Proposition 1. Specifically the variance for the complete U-statistics is proportional to 1/\sum n_k' whereas the variance for the incomplete U-statistic is 1/\prod {\choose..}. (see Eq. in Prop. 1) This difference is huge and the mathematical formulation is counter-intuitive. If we set n_k'=n_k

shouldn't the complete U-statistic be efficient like theory suggests? It seems that efficiency should depend on

terms like n_k' / n_k. This would not follow from Prop. 1. Unfortunately there is no proof for Prop. 1, but a citation that is not very helpful.

I strongly suggest that the authors provide a simple proof for that fact because it is also

important for Theorem 2, and for the paper in general.

The other major problem is mathematical clutter. For example, Def. 1 is a quarter page long and overwhelming. A more intuitive explanation of U-statistics is possible since this is a well-developed subject in statistics. I would recommend to build up from primitives, e.g., introduce U-statistics when K=1, and then

generalize. I also think that Proposition 2 and Theorem 1 are standard theory

in stochastic approximation, and should be in the Appendix. This section

confused me the most. How is normality used?

Is the purpose of this section to show that the conditional variance is important

for convergence of SGD? If it is, then the section is unnecessary because it replicates standard theory

results. If not, the authors should try to be more explicit and precise of what this section is

about and what are the new results.

The experimental results are limited and they don't seem to support

the claim "the incomplete U-estimator significantly outperforms its complete variant". For example, the results in Figure 1 show that the difference between the two methods

is not statistically significant, unless plotted standard deviations do not reflect

standard errors; in such case the authors should make this explicit.

Finally, there are several parts in the paper that I was confused or struggled through excess

wording e.g.,

1) In the abstract: "In this paper, we ague that in the large-scale setting

the inductive principle of SA for risk minimization should be implemented in a very specific manner. ...". This sentence spans 3 lines and gives no information.

2) In Lines #48-49 a description of SGD is given that is not typical. In plain SGD there is no averaging.

3) Overall the use of "tuples" is very confusing (starting from the abstract.) "Tuple" can mean many different things. Because this term is heavily used

and only becomes more clear in Page 3, I would suggest a very early definition.

4) Indexes in Line #106 of Def. 1 are non-overlapping?

5) In Line #173 it says that the complete U-statistic is "far from optimal". I think this is

evident by definition?

6) Remark 1 is confusing and could be omitted, or added at a discussion section, or

in conclusion.

7) The "conditional performance analysis" is confusing as well. In Line #207 it says that the analysis is "conditional on the observed data samples". But from what follows it seems that we simply condition on the number of samples, not the data samples themselves, is that correct?

8) In Line #215 it says "We point out that the present analysis can be extended to more general

contexts" and then the authors cite the work of Moulines and Bach on non-asymptotic

analysis of stochastic approximations. This is a very confusing statement.

What extension do the authors mean?

Minor: * Line #184 D_B should have "mathcal" * Constant C in Line #295 is used earlier,
Summary: This is a good work on a very interesting area of statistical learning. However, there are important flaws in the submitted version of the paper,

particularly in accuracy of theoretical results and significance of experimental results. Furthermore, several parts of the paper appear to be unnecessary.

Author Feedback
Author rebuttal: We thank the reviewers for their very helpful comments. Some reviewers have suggested interesting modifications in the presentation of the paper that we will take into account to improve the quality and clarity of the manuscript. Below, we answer specific questions and address some technical points raised by the reviewers, in particular a clarification of Proposition 1.

*** Assigned_Reviewer_1 ***

Although not rooted in theory, step sizes of the form \gamma_t = \frac{\gamma_1} {1+\gamma_1 \lambda t} are often used in practice in SGD for better convergence and to avoid very large steps in the beginning of the optimization. Note that the step size asymptotically decreases like \frac{1}{\lambda t}. To avoid confusion we will instead show the results for \gamma_t = \frac{\gamma_1}{\lambda t}, for which the same conclusions hold.

*** Assigned_Reviewer_2 ***

We feel that the criticism of the reviewer is a bit unfair. We do not mean to compare ourselves with specific state-of-the-art methods in clustering, ranking, metric learning, etc. Instead, we take a more general perspective by observing that the associated optimization problems are all in the form of ERM of U-statistics. When the training set is large, many methods resort to SGD-type algorithms to numerically solve these problems in practice. However, to the best of our knowledge, the impact of the tuple sampling strategy on the performance of SGD has never been carefully investigated. Our paper provides a novel analysis and argues that one strategy should clearly be preferred over the other, providing both theoretical and experimental evidence of its superior performance. We believe that our study can help practitioners implement efficient SGD algorithms to solve this general family of problems.

*** Assigned_Reviewer_3 ***

We first want to thank the reviewer for his/her several useful comments to improve the clarity of the paper.

- It seems that the reviewer slightly misunderstood the nature of the statement in Proposition 1: the variances \sigma are w.r.t. the gradient of the *true* loss L(\theta) and not that of the empirical loss \hat{L}_n(\theta), unlike the empirical variances \sigma_n of Section 3.2. We will make this more clear and provide a basic proof. While the huge difference in variance shown by Proposition 1 might sound counterintuitive at first, it is due to the fact that the incomplete U-statistic estimate involves many more different observations. For instance, take the case where K=2 and d=1, and we are given n i.i.d. observations. For a batch size B=n'(n'-1)/2, the number of different observations involved in the complete U-statistic is exactly equal to n', while for the incomplete U-statistic it is always superior to n' (more precisely, if n' is not bigger than \sqrt{n} then the expected number of different observations involved is of the order of n'(n'-1)).

- Note that Figure 1 shows averages over 50 random runs of SGD. While the performance can indeed vary quite a lot from one run to the other (which is common in SGD algorithms), on average (solid lines) the incomplete U-statistic estimate significantly outperforms the complete estimate. We also stress the fact that the case K=2, d=1 (as in the AUC optimization results of Figure 1) is the "least favorable case" for the incomplete U-statistic (as seen in Proposition 1). The results for K=1, d=3 given in Figure 2 show a much larger gap, which is consistent with our theoretical results.

- In Section 3.2, we work conditionally upon the data sample in the sense that we only deal with the empirical risk, which is defined w.r.t. the data sample.

- Finally, the kind of simple extension we have in mind is to address the smooth but not strongly convex case, for instance adapting the analysis in Bach and Moulines (NIPS 2011).